# A Comparison of the General Knowledge and Skills Displayed by Students Participating in International Geopolitical Competitions

**Piotr L. Wilczyński**

Institute of Geography, Pedagogical University of Krakow, Podchorążych 2, 30-084 Kraków, Poland; piotr.wilczynski@up.krakow.pl

**Abstract:** Three years ago, the Polish Geopolitical Society began an initiative focused on students, PHD candidates and interested academic societies, who wished to co-operate in popularizing the subject area of geopolitics. This initiative sought to serve as a forum for such groups and individuals to compete with other interested colleagues and groups from around the world. During the process, students strive to prove their level of professional knowledge, while their teachers assist them in preparing for their best presentations. All participants then meet during the final stages of the competition in order to exchange their experiences. This, in turn, benefits the development of general approaches and methods of study regarding the discipline of geopolitics. The question addressed in this paper then, is how such international competitions can improve the overall skills and knowledge of the subject area at hand among those participating. The importance of this question is underscored by various initiatives undertaken that attempt to measure the quality of higher education. The research presented in this article, then, is based upon interviews with both participants and organizing committee members, which attempt to gauge the experiences and results achieved during such competitions. The results show both the positive and negative aspects of organizing such gatherings. Most certainly, one could draw the conclusion that such events are the most attractive to the most ambitious of students and teachers, who consider education a privilege and as a process, which continues throughout one's lifetime. Adversely, for those who place education in the same category as a material good, to be bought and sold, such competitions have little appeal especially when focused upon a narrow field of study.

**Keywords:** education quality; international cooperation; competition; geopolitics; teaching methods; educational initiative; educational events

## 1. Introduction

Improvements in the quality of service to customers constitutes one of the main goals of healthy businesses the world over. Universities and the education they offer are not an exception to this rule. As universities compete to attract more funding, prestige and of course students, students themselves enter into competition with one another in order to attain higher degrees and the skills that accompany them. Y.C. Cheng and W.M. Tam in their publications note that in the mid-1990s a strong emphasis was placed internationally on educational reforms. They both present models as to how these efforts succeeded or failed [1]. Their work serves as an example of the "mathematical modeling approach" in educational science so very popular during those years. A much different approach was used by other economists who treated education as a tool for the economic development of various countries. These economists treated the university as a sort of "academic factory" whose main task was to produce skilled labor. Education understood in this context then, is to serve as a stable commodity thanks to which the

economic conditions of nations (including productivity and efficiency) can be improved. This approach is frequently referenced, for example, in publications of the World Bank [2]. Worth noting also is what is called the "institutional approach". H. Fabrice examines this approach in his writings [3]. This article is based mainly upon Fabrice's approach, which sees education and its effectiveness as taking into account a wide range of defined methodologies, perspectives and education research.

It goes without saying that there exist various ideological perspectives that inevitably inject themselves into the study of the quality of education. For example, presently, it is very popular to examine the many aspects of the quality of education through the lens of gender and social justice or through the microscope of critical science. Such approaches are very popular, since they create many controversies, resulting in such views being cited in a variety of educational journals [4].

This paper also sheds light on the fruits of international cooperation among various academic centers as gleaned from their student's participation in the International Geopolitical Competition, as well as from the participating staff members involved in the preparations for the competition. Of great value are the preparatory meetings held in December, when representatives meet to exchange views and experiences gained during previous, yearly conferences as well as to discuss the results of the competition in the form of student awards and accomplishments.

The recognition the competition participants bring home is also evaluated in this paper. It can surely be stated that the International Geopolitical Competition has a strong international component focused on fostering international cooperation between the various educational institutions represented. Although the competition is organized by the Polish Geopolitical Society, our yearly competition has witnessed the steady growth of international participants [5]. This is thanks to the already existing international connections the Society maintains with other educational entities beyond the borders of Poland. It is, however, worth noting that the Polish Geopolitical Society, unlike other such societies, is an independent and self-functioning think-tank and receives no public funds. Most of education international cooperation research papers are commonly focused on public funds for internationalization policies [6].

There is some literature about international competitions for students, but not so popular as research subject like quality of education or international cooperation mentioned above. Nevertheless, globalization and international competitions are becoming more and more popular around the world of higher education. Research into the role international cooperation plays in the growth and development of international competitions has been practically null and yet the popularity of such competitions in the world of higher education cannot be ignored [7]. They are especially being organized in the fields of ecology [8], science and engineering [9], international law [10], mathematics and computer science [11], as well as in the areas of astronomy and astrophysics. International competitions in geopolitics and political science are, however, more of a rarity. When they do occur, they are usually not of an international nature. In this context, the Polish Geopolitical Society fills a certain void by organizing its yearly competition. [12,13].

Universities famous for publishing rankings such as the Shanghai, Leiden and heavily cited Wikipedia Rankings all use the "weight and sum" approach in order to process data. Although this approach would seem to pass the common sense test, it has its statistical complications [14]. Unfortunately, none of these rankings take into account the results of international educational competitions. Perhaps it would be advisable, but then again, only a handful of universities hold such competitions and arriving at standards and methods by which to rank them would be difficult. Nevertheless, the idea is worthy of consideration. It would be interesting to see which universities would come out on top as they underwent ranking.

Research conducted during the past three International Geopolitical Competitions was focused primarily on one main question, that is: how do international competitions improve the quality of higher education. To help answer this question is the main goal of this paper. Taken into account while preparing it were basic descriptions of the general educational quality of the competitions held, expressed international criteria in organizing them, as well as university rankings. The groups

represented in our yearly competitions were composed of high-school students, students and the university level, PHD students and recent graduates. This paper will not focus on any one group, but instead will treat the participants as a whole, examining their educational experience in the context of a continual, life-long process of learning, and not that of one particular age group or educational level. To help answer the primary and other questions, interviews were conducted with members of the board, organizing committee, participants and their teachers. Over 40 interviews were conducted. While the same questions were asked of all respondents, they were not required to answer them all. They were all encouraged via open-ended questions and an informal, conversational tone from the interviewer to discuss the subject as they wished. All recorded interviews are archived in the headquarters of the Polish Geopolitical Society in Kraków.

These were semi-structured interviews, focused on perceptions and opinions of the competition. The questions were open ended and they were not asked in exactly the same way or in exactly the same order to each and every respondent. However, all interviewees were asked about the following:

- Why did you engage yourself in the competition?
- How did you get an information about the competition?
- What are your goals in taking part as participant/organizing committee/jury?
- What are disadvantages and benefits of such competitions in your opinion?
- What problems did you encounter during preparation and tests?
- What are your general concerns about the competition?
- What could be improved in next year edition?

With such in-depth interviews, the primary aim is to hear from respondents what they think is important about the topic at hand and to hear it in their own words. To collect and preserve the information provided by participants, audio recordings were made and fields were made. This research presents and analyzes the participants' stories in their own words as well as others'. Here, we presented and analyzed participants' stories, as well as other gathered data. A thematic analysis of the interviews was conducted and examples of the most significant responses are included as examples in the results section.

## 2. IGC—Editions, Rules, Organizers

The International Geopolitical Competition serves as the source event for this paper's analytical and empirical research since the IGC exemplifies the aforementioned trends in education. The phenomena of globalization and international cooperation have served as the springboard for new competitions that popularize the sciences, contributing to an improvement in the quality of education. This year marks the third International Geopolitical Competition characterized by an increase in the number of participants, especially from the geographical areas of Central and Eastern Europe. This increase in interest proves not only the growing popularity of geopolitics as a discipline of study, but in turn defies the present COVID-19 narratives. Research clearly shows that the correlation between the epidemic and participation levels in educational competitions has not resulted in a decrease in the level of participation, but instead in an increase.

Contained in the first printed edition of the Polish Geopolitical Society's goals, the management of the Society along with a speech by its newly elected director [15], outlined its vision and goals for the future development of the organization so the very first pilot edition was tried to begin in 2014. Seven such goals were put forth. The first goal mentioned was to assist the Society's members in their own scientific development. Secondly, the necessity of independence from governmental or foreign influence was underscored. The third goal mentioned defends the neoclassical perspective of geopolitics as an academic discipline. This is important, since geopolitics is still treated by many professors from post-communist countries as a "fascist ideology" [16–18]. The fourth goal accents the importance of the Scientific Council, one of the Polish Geopolitical Society's main bodies, which is designed to defend deliberation and the exchange of knowledge and ideas against ideological infiltration.

The fifth and sixth goals touch upon the role the Society in general plays within the context of shared geopolitical interests. The seventh and last goal of the Society focuses upon the promotion of the geopolitical sciences and the role IGC may play in this process.

Unfortunately, following the crystallization of goals, the Society encountered some obstacles. An interview with J. Dutka (Appendix A), a member of the board at that time, reveals that the main challenge facing the Polish Geopolitical Society plans was not only the lack of funds, but also a certain disapproval and lack of any support by the Ministry of Science and Higher Education. Moreover, tensions between the Society's chairman and vice-chairman prevented the Society from making any notable progress. A new board was elected in 2017 for the next three-year term. The examination of such experiences of the Society could serve to prevent other organizations from abandoning their efforts and plans when faced with similar challenges.

Finally, the first edition of IGC took place in 2018 [12]. Before that, members of the board of Polish Geopolitical Society have agreed about all needed documents, schedules, and other important things, such as IGC Resolution, the Terms and Conditions of Participation, and the composition of the Organizing Committee [19,20]. Details concerning their publication are provided by J. Trubalska, a member of the board from 2017–2020 term (see: Appendix A).

As already mentioned, the goal of the International Geopolitical Competition is to attract the very best student/participants possible, honoring their display of knowledge with honorable mention as well as prizes. Both Polish and English serve as official languages during the competition, and are correlated at the same difficulty level. Students from all educational levels are invited to apply, the only stipulation being that they not hold a PHD at the time of sign-in. Registration from the competition is, however, very simple. Foreign participants need only fill out a registration form. Along with the registration form, Polish participants pay a small fee [19]. All participants along with their sponsors are expected to cover all other costs connected with their participation during the competition.

The competition is held in three stages. The first stage can be dubbed as the Qualifying Stage. The second consists of participation in a regional competition, while the third and final stage consists of participation in the main competition itself. In addition, there are three governing bodies at work during the competition. They consist of the Organizing Committee, the International Professor's Board and the Revisions Committee. The Organization Committee is established by the competition's chairman and fulfills an organizational and marketing role. It supervises activities during the competition, for example, the preparation and correction of tests. The International Professor's Board serves in an advisory capacity as well as serving as judges during the competition's final stages by reviewing essays and test answers. They may also disqualify participants caught cheating in these areas. In addition, the International Professor's Board plays an important role in promoting the IGC in the international, academic world. Finally, the Revisions Commission is responsible for hearing all appeals when misunderstandings occur, making final judgments in such cases [19,20].

As previously mentioned, the IGC is comprised of three stages. The assessment criteria for all three stages are available to all participants [20]. The first stage begins after the best of participants is established by the Organizing Committee. This occurs in May. The main goal of the first stage is to eliminate those participants who do not possess sufficient skill levels. Some participants may be able to skip this first, qualifying stage. They are comprised of (A) those who have published reviewed, scholarly books or articles in scientific, peer-reviewed journals on topics included in the competition. Such participants need only send a file or link to their articles for review; (B) officers and NCOs of all branches of the military services, police or students of military academies with active military service ongoing. They need to send a proper document scan to prove their right; (C) participants of previous editions final stages and those participants who the chairman himself promotes as a matter of exception, who are specialists in the area of geopolitics and who may add weight and prestige to the quality of our competition. If a participant does not belong to one of these three exceptions, he or she must take part in the first stage of the competition by writing an essay composed of 10–12 pages on a contemporary geopolitical event such as border skirmishes, battles, new conflicts

or the like. There are in place strict guidelines for assessing these essays. These criteria help to assess the participant's editing skills, (with a scoring range from −10 to 10 points) and essential skills, (with a scoring range of −40 to 30 points). Assessment criteria are part of the Terms and Conditions of the IGC., so all participants know them before start. The maximum number of points which may be earned is 40. Unfortunately, those who receive a negative grade, or who do not submit their essays on time, cannot be admitted to the next stage of the competition. Essays are graded by two individual reviewers, according to the precepts of "double peer-review". The final grade established is the result of the average sum of points from the two independent reviewers. The best of the essays are then published without charge in the scientific journals of the Polish Geopolitical Society as a prize for such first-stage participants. The scientific journals of the Polish Geopolitical Society are: The Geopolitical Review and The European Journal of Geopolitics [19,20].

The second stage of the IGC is held in the headquarters of each regional unit (Figure 1), or at an established location within other countries. Usually, these countries already have an established relationship with the Polish Geopolitical Society. In some cases, for foreigners, possibilities exist to take part online. Participants who qualify for the second stage of the IGC must pass a test. The test is comprised of 60 questions, covering various sub-disciplines in the area of geopolitics. Each question is provided with six possible answers, five being false and one being correct. During the test, the participant is able to use any sources, excluding consultation with other participants or staff. The participants may leave the test area only after the competition of the test itself. It is forbidden to use mobile devices or telephones or to copy answers from others. Any sort of cheating is met with a score of zero and the offender is disqualified from continuing in the competition. All participants should have their IDs with them as they enter the testing site and late-comers will not be allowed into the room. Both the questions and the multiple-choice answers for the test are rendered in Polish and in English. A participant gains 1 point for each correct answer, 0 points for unmarked answers and −1 for incorrect answers. The test begins at the same hour everywhere it is being held and the questions are exactly the same regardless of the location. There are 2nd-stage test questions, and the correct answers are available online. All the questions are prepared by the Scientific Board of Polish Goepolitical Society. These professors are sign off each question. The results of the test are posted on IGC webpage the same day of the competition. The jury has a mark scheme so that checking and marking the test is a quick procedure. The 20 best scorers gain entry into the final stage of the competition [20].

Teachers and students as well are encouraged to share their experiences and comments by using an online forum provided by organizers after second stage [21]. In addition to yearly competition, the Polish Geopolitical Society has plans to establish summer camps which would also add to an increase in the educational quality as concerns geopolitical studies, as well as an increase in interest in the subject area and internationalization.

The final stages of the IGC provide the most emotion, heightening the overall educational experience for both participants and their teachers as well. This is confirmed in a post-competition interview with N. Adamczyk, M. Sima and P. Nawała. This final part of the competition is held a day before the annual meeting of the Polish Geographical Society, which takes place at various sites in Poland on the first weekend of December. In 2018, the annual meeting was held at the Warmia-Mazury University of Olsztyn [22] and in 2019 at the University of Lesser Poland in Oświęcim [23]. The main, annual meeting for 2020 is planned for the State University of Chełm in 2020 [24]. The final stage of the competition is conducted in three parts: (a) in written test form; (b) orally; and (c) in the form of practical exercises. The written test of this stage resembles that of the previous stage's written test, but the difficulty level is much higher. The oral test differs greatly in style, with participants choosing questions randomly from a bag. They then must formulate their answers on-the-spot. Their answers are judged by a jury of three professors with scores ranging from 0 (for no answer or for an incorrect answer) to 3 (for excellent answers). In addition, participants are asked to work with a map, and answer questions in order to support their sphere of influence on this map. This involves the proper

application of the geographical knowledge and skills of each participant. The participants present their understanding of spheres of influence, war, non-aggression pacts, alliances and peace treaties. Since this is all done in the company of a group of participants, each participant is able to gain points, eliminate competitors and even choose questions for their competitors to answer. The importance of such interactive, decision-making "games" or exercises is emphasized in the research of C. Czauderna and A. Burke not long ago [25]. The third part of the final stage is comprised of engaging in a more specific strategic exercise prepared by the Organizing Committee, which draws upon the geostrategic knowledge of each participant. This exercise relies on the same basic rules employed in the preceding general strategic exercise, but concentrates on accepted strategies used by armies from around the world. Since military skills as well as knowledge play a role in geopolitics, participants undertake the roles of generals in command of brigade level military units groups. The participants then are tasked with various military objectives and achieve them on what resembles a board game, thus proving their practical knowledge on the strategic, operational and tactical levels. Breaks of one hour occur between each of the three parts of this stage of the competition. All three parts of this stage are scored according to the set standard, with the highest achieving participants earning 10 points, which earns first place status, nine points—second place status, etc.—with the student in 10th place receiving only 1 point. The remaining half of the 20 participants do not qualify for an award. So then, the highest score possible is 30 points and the lowest is 0 points. The three participants with the highest scores are declared winners and are awarded with gold medals. Those with 10 points or more receive silver medals and those earning more than three points are given bronze medals. This is in addition to other more valuable prizes provided by sponsors. All results are publicly announced on the following day during the opening celebrations of the Annual Meeting of the Congress of Polish Geopoliticians [26], during which all prizes and medals are awarded.

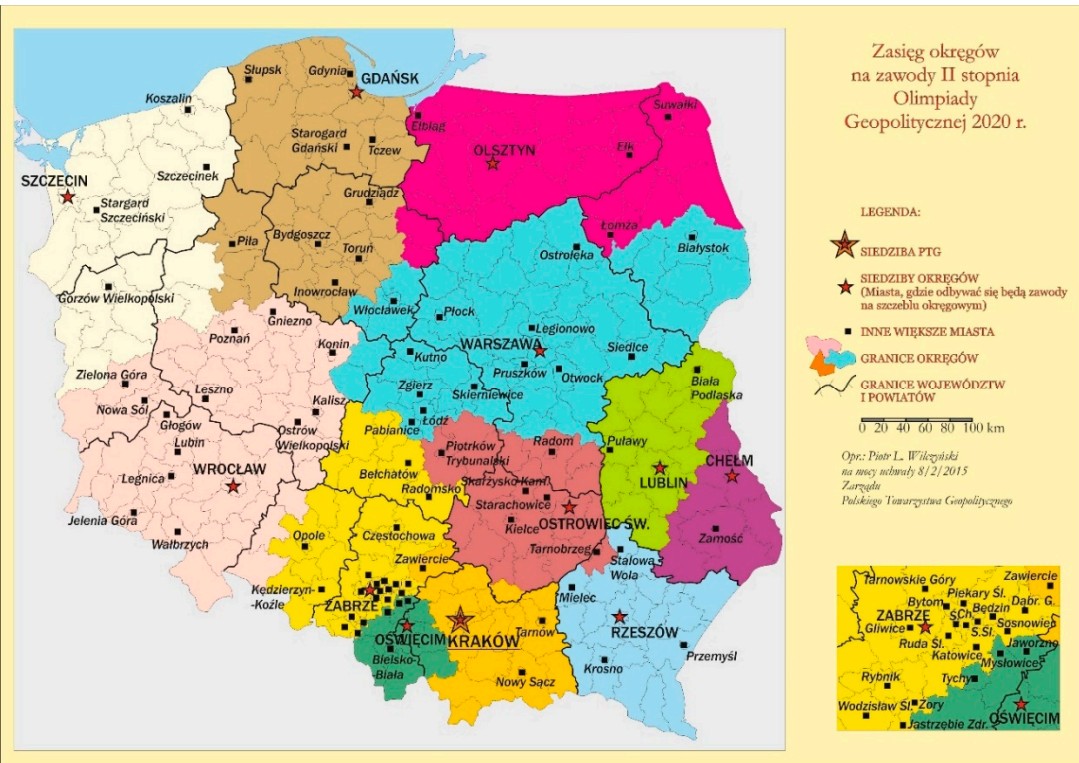

**Figure 1.** Headquarters of the Polish Geopolitical Society's regional units (stars) and their territorial range for 2nd-stage IGC participants (different colors) on an administrative map of Poland for the 2020 edition. Source: https://ptg.edu.pl/3-edycja-2020/ (accessed on 7 May 2020).

## 3. Results

As already mentioned in this article, all participants in the IGC are divided into three categories. The first category is composed of the youngest participants who are of high school age. Members from this group do not usually pass to the more advanced stages of the competition, but their participation serves as valuable experience for participating in such international competitions and serves as motivation for better achievement in their high school curriculum of studies. These students often put their all into composing the best of qualifying essays during the first stage (as said an experienced high school teacher J. Dutka in an interview). Most of the remaining participants of the competition represent the second group, which is composed of bachelor and master studies level university students. Many of them hope to earn valuable points in order to successfully apply for scholarships, but the idea of competing in final stage of an international competition accompanying geopolitical conference serves as the strongest motivation for their decision to enter the IGC (as said participants: P. Nawała, Z. Chechliński, M. Sima, A. Borówka). The personal impressions of some finalists of the IGC can be found on the Polish Geopolitical Society's Youtube channel, available in Polish [27]. Those comprising the third group of the competition are generally PhD candidates and recent university graduates. Their motivation for taking part in the competition is to display their knowledge, passion and command of subject matter, as well as to meet others who share their interests (as said A. Myślicki). While high school and university students do not generally pass on to the last stage of the competition, those comprising the third group do. Data supporting this is shown in Tables 1–3 [28].

**Table 1.** Participants of each IGC 2018 stage divided by their origin and level of education (H—high school students, U—university students, D—PhD candidates and graduates) [28].

| Origin by Territory | IGC 2018 Participants of Each Stage and Winners | | | | | | | | | | | | | | | | | |
| | I Stage | | | II Stage | | | III Stage | | | Bronze Award | | | Silver Award | | | Gold Award | | |
| | H | U | D | H | U | D | H | U | D | H | U | D | H | U | D | H | U | D |
| Foreign | - | 5 | 2 | - | 1 | 1 | - | 1 | 1 | - | 1 | - | - | - | - | - | - | - |
| Gdańsk | 4 | 1 | 1 | - | 1 | 1 | - | 1 | 1 | - | - | - | - | 1 | - | - | - | - |
| Kraków | 5 | 7 | 2 | - | 4 | 2 | - | 4 | 1 | - | 1 | - | - | 2 | - | - | - | - |
| Lublin | 5 | 1 | - | 4 | 1 | - | - | 1 | - | - | 1 | - | - | - | - | - | - | - |
| Olsztyn | 5 | 1 | - | 2 | 1 | - | - | 1 | - | - | - | - | - | 1 | - | - | - | - |
| Rzeszów | 10 | - | - | 7 | - | - | 1 | - | - | - | - | - | - | - | - | - | - | - |
| Szczecin | 5 | - | - | 4 | - | - | 1 | - | - | - | - | - | - | - | - | - | - | - |
| Warszawa | 7 | 2 | 2 | 1 | 1 | 2 | 1 | 1 | 1 | - | 1 | - | - | - | - | - | - | 1 |
| Wrocław | 5 | 3 | 1 | 2 | 1 | 1 | 1 | 1 | 1 | - | - | - | - | - | - | - | 1 | 1 |
| Zabrze | 7 | 3 | 1 | 1 | - | 1 | - | - | 1 | - | - | - | - | - | 1 | - | - | - |
| total | 53 | 24 | 9 | 21 | 10 | 8 | 4 | 10 | 6 | 0 | 4 | 0 | 0 | 4 | 1 | 0 | 1 | 2 |
| | 86 | | | 39 | | | 20 | | | 4 | | | 5 | | | 3 | | |

**Table 2.** Participants of each IGC 2019 stage divided by their origin and level of education (H—high school students, U—university students, D—PhD candidates and graduates) [28].

| Origin by Territory | IGC 2019 Participants of Each Stage and Winners | | | | | | | | | | | | | | | | | |
| | I Stage | | | II Stage | | | III Stage | | | Bronze Award | | | Silver Award | | | Gold Award | | |
| | H | U | D | H | U | D | H | U | D | H | U | D | H | U | D | H | U | D |
| Foreign | - | 6 | - | - | 5 | - | - | 5 | - | - | - | - | - | - | - | - | 2 | - |
| Chełm | 2 | 2 | - | - | 2 | - | - | 1 | - | - | - | - | - | 1 | - | - | - | - |
| Gdańsk | 5 | 3 | 2 | 2 | 2 | 2 | - | 2 | - | - | 1 | - | - | - | - | - | - | - |
| Kraków | 8 | 3 | 3 | 6 | 3 | 2 | - | 2 | 2 | - | 1 | 1 | - | - | 1 | - | - | - |
| Lublin | 1 | 1 | - | - | 1 | - | - | 1 | - | - | - | - | - | 1 | - | - | - | - |
| Olsztyn | 2 | - | - | 2 | - | - | 1 | - | - | - | - | - | - | - | - | - | - | - |
| Rzeszów | 1 | - | 1 | 1 | - | 1 | - | - | - | - | - | - | - | - | - | - | - | - |
| Szczecin | 2 | - | - | 2 | - | - | 2 | - | - | - | - | - | - | - | - | - | - | - |
| Warszawa | 6 | 1 | 2 | 4 | 1 | 2 | - | 1 | 2 | - | - | - | - | - | 1 | - | - | - |
| Wrocław | 9 | 1 | 1 | 3 | - | 1 | - | - | 1 | - | - | - | - | - | - | - | - | 1 |
| Zabrze | 10 | - | - | 8 | - | - | - | - | - | - | - | - | - | - | - | - | - | - |
| total | 46 | 17 | 9 | 28 | 14 | 8 | 3 | 12 | 5 | 0 | 2 | 1 | 0 | 2 | 2 | 0 | 2 | 1 |
| | 72 | | | 50 | | | 20 | | | 3 | | | 4 | | | 3 | | |

**Table 3.** Number of participants of IGC 2020 first and second stage divided by their origin and level of education (H—high school students, U—university students, D—PhD candidates and graduates) [28].

| IGC 2020 Participants of I and II Stage (Finalists not Known Yet—Competition is Ongoing) | | | | | | |
|---|---|---|---|---|---|---|
| Origin by Territory | I Stage | | | II Stage | | |
| | H | U | D | H | U | D |
| Foreign | 2 | 27 | 19 | - | 14 | 19 |
| Chełm | 7 | 4 | - | 5 | 4 | - |
| Gdańsk | 1 | 3 | - | - | 1 | - |
| Kraków | 5 | 6 | 5 | - | 5 | 3 |
| Lublin | 7 | 1 | - | 3 | 1 | - |
| Olsztyn | 1 | - | 3 | - | - | 2 |
| Ostrowiec | 1 | 1 | - | - | - | - |
| Oświęcim | 1 | 1 | - | - | - | - |
| Rzeszów | 5 | - | - | 3 | - | - |
| Szczecin | 2 | - | - | 1 | - | - |
| Warszawa | 4 | 1 | 2 | - | - | 2 |
| Wrocław | 4 | 7 | 5 | - | 4 | 4 |
| Zabrze | 7 | 1 | - | 1 | 1 | - |
| total | 47 | 52 | 34 | 13 | 30 | 30 |
| | | 133 | | | 73 | |

The first IGC held in 2018 drew a total of 86 participants, including seven from abroad. The international participation rate was rather low, since funding was limited for actual marketing of the competition. Countries represented in that year were mostly from Central Europe and others such as: Azerbaijan, Belarus, the Czech Republic, Hungary, Kazakhstan, Turkey and the United Kingdom. Of the total group, 39 participants passed on to the second stage of the competition, with one participant from the UK and Kazakhstan each advancing. These individuals also passed on to the third stage to which only 20 participants are advanced. The results with the international achievements are shown in Table 1. Only 12 of the 20 participants of the final stage received awards at the conclusion of the 2018 competition.

The hopes were that the number of participants for the 2019 IGC would grow, but because of a lack of funds, the Polish Geopolitical Society was not able to promote the competition properly. Only 72 participants took part, with six foreigners from Belarus, Germany, Great Britain, Hungary, and the Ukraine. So again the international participation was below 10%. Steps were then undertaken to try to increase the level of international participation in the IGC (as said N. Adamczyk). Interestingly, the overall skills of the participants for the 2019 competition exceeded those of the previous year. Out of a total of 72 participants, 50 were advanced to the competition's second stage. This is all the more worthy of noting, since both the essay topics and general questions were not of lesser difficulty. Five of the six participants from abroad were advanced to the third stage, with two of them receiving gold medals. The results of international participants are shown on Table 2. Just 10 of the 20 finalists won awards at the 2019 competition.

At the time of this article's writing, we have only received the results of the 2020 competition's first stage. Endeavors to increase international participation for the 2020 competition were, however, very successful, with 48 out 133 participants being from other countries. Foreign participants represented far more than the countries of Central Europe. Foreign participants were from: India (6), Belgium (4), Bangladesh, Brazil, UK, Ukraine, USA (3 each), Nepal (2) and Argentina, Australia, Bolivia, Bosnia and Herzegovina, China, Dominican Rep., El Salvador, Germany, Hungary, Kazakhstan, Lithuania, Mauritania, Montenegro, Netherlands, Norway, Pakistan, Romania, Russia, Senegal, Sweden, Turkey (1 each). Seventy-three participants passed to the second stage of the competition, including 33 participants from abroad (see Table 3).

While not all participants were interviewed to address the research questions, more than 40 interviews were conducted to collect the views of teachers and staff. After the transcription and analysis of the interviews, there were some noticeable suggestions and results. The first question (why did you participate in the competition?) was answered differently by high school students, university students, PhD candidates and those who had finished their studies but are still interested in geopolitics. This paper's results section describes the main motivations for participating. With the second question (how did you get information about competitions?), the majority of participants were recruited by their teachers, though some also responded to an announcement on the society's webpage and social media. Third-question (what are your goals in taking part as a participant/organizing committee/jury?) personal goals seem to be different for each person depending which group they are members of, competition participants or others. However, in many cases, this interview question provided interesting qualitative data. The aim was to allow participants to have some control over which or to what extent various topics were discussed, and this question in particular showed a variety of answers, as discussed below. The next question (what are disadvantages and benefits of such competitions in your opinion?) was also very important. Many expressed strong opinions on the benefits and disadvantages of such competitions. Many answers were concerned with the technical and financial aspects, but other noticeable and worth mentioning opinions were gathered. These will be utilized by the Polish Geopolitical Society to improve future competitions.

After processing and transcription, there were no significant or exceptional answers to the next question: what problems did you encounter during preparation and the tests? In fact, many skipped it. It is possible that is was poorly formulated or asked in a wrong way to collect any significant data. Supposedly, there were insufficient explanation for conducting the research with this question. Two last questions were very important, and some conclusions from them are detailed below. We asked what are interviewees' general concerns about the competition and what could be improved in the next year's event.

## 4. Discussion and Conclusions

Since no review of previous IGC competitions were conducted, one can only refer the reader to certain the impressions of the competition offered by participants [8,9,11]. It can generally be stated, however, that all IGC competitions have been well organized, with no serious complaints from participants or teachers. All of what was promised by the organizers was delivered upon by the competitions Organizing Committee members.

After presently analyzing collected data through the help of questionnaires and in-person interviews, it is possible to accentuate certain, viable, positive and negative aspects of the IGC and its role as a tool for educational enrichment. The level of engagement necessary for participation in the IGC is very intense. Much preparation is needed in order to participate, consuming both time and energy, in order to reach the competition's final stages. Only the most ambitious students of geopolitics would consider entering. These statements are echoed by both teachers and students during the aforementioned interviews. Since rather low participation rates in such a competition cannot claim to wield wide influence on the improvement of educational standards as a whole, most governmental agencies seem uninterested in co-funding such an endeavor. On the other hand, the excitement level of those interviewed and who advanced to the final stages of the competition cannot be denied, as, almost always, they decided to return and participate once again. Those interviewed also attest to the fact of improving their knowledge and level of skills because of their participation, a trend that can be confirmed by repeat participant's rising scores with each consecutive competition. The question can be posed whether all of the effort expended by the Organizing Committee, together with all of the personnel and spent funds is worth it. On the positive side, aside from the great costs in time, personnel and spent funds, the IGC, through its efforts, is able to help foster the educational development of young, enthusiastic devotees of geopolitical studies, while at the same time maintaining contact with them. This is most in keeping with the Polish Geopolitical Society's statutes, which should result in an increase in governmental funding. This depends, however, on the management skills of each specific, organizing entity, but it goes without saying that well-respected and well-organized

international competitions, in various areas of education, could serve as a basis for organizing institutions to receive additional funding. There is a debate as to whether high school students should participate at all in the competition. Some voices feel that combining this age group with doctoral candidates is a waste of time and resources. However, others point out that the area of geopolitical studies is very weak in most educational institutions and that combining these various age groups only serves to improve the situation at large. In addition, we should note that, according to IGC statistics, some exceptional participants from the younger age groups have actually advanced to the final stages of the competition, while some at the doctoral level participants were eliminated (Tables 1 and 2). Of course, this is more the exception than the rule, but this fact lends support in favor of continuing high school age participation in the IGC.

Also worth noting is the increase of international participants in this year's competition. The data have not yet been gathered in full, but this noticeable trend should most certainly have a positive impact on the quality of education fostered by the competition, as both teachers and students share their knowledge and experience in an international setting. The organizing committee seeks to ever increase the level of international participation in the competition. It is worth mentioning that the interviews which have been conducted with foreign participants show that the most enthusiasm was garnered by the very final stage of the competition, with all of its accompanying events.

The interviews conducted with competition participants serve as an effective good tool for gathering opinions (both the good and bad), concerning the rationale for organizing such educational events. In an interviews, the most positive aspect of our competition mentioned was the ability to meet renowned authorities in the area of geopolitics, as well as others who share the same interests. Another aspect much appreciated by participants was the division of the competition into three stages. It is felt that this makes the competition more interesting and engaging. Most of those interviewed found the venue to be attractive for the comparison of knowledge and skills. Also much appreciated was the well-defined field of knowledge. The questions were of a high standard, often pointing to aspects of geopolitical studies, which are in need of more attention. This is due to the well-described literature section in the Terms and Conditions of the IGC [20]. No doubts remain that the IGC could be even more attractive if the Polish Geopolitical Society could afford to organize more events.

Aside from the positives mentioned in the interviews with participants, the following negatives were also mentioned. Firstly, the prizes offered were of a lower value than most other international competitions. Secondly, some complaints were registered regarding the level of difficulty of some of the questions. In response to the level of difficulty, one can only say that the questions prepared by professors were the same for all participants during the second stage of the competition; moreover, the selection of questions from the "bag" had the same random risk for each participant. Thirdly, some participants complained of computer glitches. This seemed to have affected the participants of the third stage the most. Unfortunately, this was due mostly to a lack of funding. Fourthly, many participants of the third stage complained that too many activities were packed into one day, causing fatigue. It was suggested that the exercises of the third stage be spread out over two or even three days. This extra time they noted was necessary for all in order to absorb and process new information and knowledge. The last negative aspect of final competition mentioned was the lack of free time.

The above-mentioned, constructive criticisms will be taken into consideration while preparations for future competitions will go on. Although organizational efforts are great, organizers feel that it is well worth promoting such an international competition, which can, in a certain sense, serve to help increase the level of knowledge, enthusiasm and passion for geopolitical studies among participants, often more so than within the confines of a university lecture hall or high school classroom. Taking all suggestions into account, some reforms will be undertaken in organizing of future International Geopolitical Competitions.

**Funding:** This research received no external funding.

**Acknowledgments:** An author would like to thank Polish Geopolitical Society for support and providing interviews opportunities.

**Conflicts of Interest:** The author declares no conflict of interest.

## Appendix A

List of interviewed people. Knowledge and any information gathered during an interviews can be confirmed by enlisted man via a personal e-mails shown here, or through contact with Polish Geopolitical Society. This list is composed in order of appearance in main text.

J. Dutka, former member of the board—ptg@ptg.edu.pl (the main Polish Geopolitical Society contact e-mail).

J. Trubalska, former member of the board—ptg@ptg.edu.pl

N. Adamczyk, member of the board, Organizing Committee member—@post.diplomats.pl

M. Sima, participant and laureate of IGC 2019 from Hungary—simamaark@gmail.com

P. Nawała, participant and finalist of IGC 2018 and 2019 from Poland—piter890@interia.pl

Z. Chechliński, participant and laureate of IGC 2018 and 2019 from UK—zbigniew.chechlinski@gmail.com

A. Borówka, participant and laureate of IGC 2018 and 2019 from Poland—aleksy.borowka@gmail.com

A. Myślicki, participant of IGC 2019 from Poland—ptg@ptg.edu.pl

There were more interviews conducted (more than 40) than cited.

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
