# Peer review of "A Comparison of the General Knowledge and Skills Displayed by Students Participating in International Geopolitical Competitions"

_education, doi:10.3390/educsci10090255_

Round 1

Reviewer 1 Report

The paper opens with some excellent positioning and some interesting observations on a fairly unusual competition.

I would like to see more examples of the questions (and responses) and particularly more detail on how responses are assessed. Is a mark scheme used? Are the assessment criteria available to potential participants?

While the findings are drawn from interviews, the paper lacks any direct quotations from the interviews and does not describe the interview methodology.

I would also be interested in knowing if there was any formal feedback and evaluation process and did it gather responses from the participants.

Overall, this is an interesting and possibly unique event that deserves more coverage. However, at present the paper lacks detail to support the conclusions.

Author Response

Thank You for revising my paper. I found all comments as creative and I eagerly improved an article with:
1) Showing questions and answers examples of the competition. They are available online so I put a reference to them (ref. 29 and 30). I added some text about it on a pages 4-5. All assesment criteria are available for participants on Society's web page (ref. 20). The jury uses a mark scheme to quickly check written tests and essays quality.
These test and answers are available here:
and answers: https://ptg.edu.pl/wp-content/uploads/W%C5%82a%C5%9Bciwe-odpowiedzi-etap-II.doc

2) I also added a description of an interview method and their evaluation, but direct quotations are not possible now, due to CoViD-19 situation. All interviews are archived in the Society HQ, and I have no easy access to them now. I coudn't get there within 10 days given to me by the editor for my reply. Some impressions of participants are recorded and available on Society's Youtube Channel in Polish only (ref. 27). I hope its enough, if not, I will need more time. All HQ staff is on vacation now, and there are still some restrictions due to CoViD.

I would like to admit that text was once again revised by american-english native speaker.

Thank you very much for your revision

Reviewer 2 Report

Overall, this is an interesting manuscript which provides an overview of an international competition organized by the Polish Geopolitical Society. I have one concern, the question (how such international competitions can improve the overall skills and knowledge of the subject area at hand among those participating) is not adequately answered using data from the interviews (more than 40) conducted in undertaking the study.

It would be important to report the analysis strategies used to analyze the transcript data, report on themes and findings that arose from such analysis. As it stands, there is no indication that the interview data was transcribed or analyzed, just that interviews were conducted. 

Author Response

Thank you for your revision. I noticed the lack of proper description of interview proceeding, and second reviewer mentioned it as well. That is why I found it most important thing that have to be improved.

1) Firstly i put more information in the introduction section about these interviews. There are questions as well as scope and all proceedings with data. Unfortunately I couldn't get quotations from those interviews these days due to summer vacation season and CoViD situation. There is noone in Polish Geopolitical Society HQ during this 10 days given to me for reply by editor. Nevertheless I improved description of research design sufficiently I hope, with more than one paragraph in the introduction. There are mentioned interview strategies and analysis method of transcripted data.

2) I also added results description to the "results" chapter, what might sufficiently report an overall findings of conducted interviews.

In overall a text is now longer by 2 pages, but seems to cover all reviewers requests.

I also gave this text to english native speaker from the USA, to check it again.

Thank you for your revision. I know it can be improved more, but I had only 10 days for that. I hope it will be accepted this time, and my changes are sufficient.

Round 2

Reviewer 1 Report

It is unfortunate that the interview transcripts are not available as quotations from them would add to the richness of the data and support the analysis and conclusions. However, if including them would require a significant delay then I suggest the paper be published and at a later date an appendix of interview analysis could be added.

The amendments made by the author address my earlier comments, however the English needs editing. I have proposed some edits in the attached document.

Author Response

Thank you for your suggestions. Text was checked twice by native speaker from the USA. I thought everything would be clear. I am happy you got involved in this paper.

Those interviews are still archived, and they are also only in Polish. It is still unaffordable for me. I dont know what will be happening with this covid disaster, but for this days it is impossible to predict when HQ will be reopened.

Thank you

Reviewer 2 Report

This is an improved version. The methods are now more fulsome. While it would have been good to have presented evidence from the transcripts to show the types of responses that the researchers received, the explanations presented are sufficient for this paper. However, the repetition of the interview questions in lines 339-343 (also noted in lines 111-119) are redundant. 

While the author/s indicate they had the paper proofread for English, there are a number of significant grammatical errors in the edits. This paper will need to be further edited for English grammar. 

Author Response

Thank you for your revision. The text have undergone language check again with another native speaker, and some changes have appeared. All errors should be solved now. I have changed redundant part following your coment. Thank you